# Effect of freeze–thaw cycles on root–Soil composite mechanical properties and slope stability

Ruihong Wang[1,2], Zexin Jing[2], Hao Luo[2]*, Shun Bao[2], Jingru Jia[2], Xiaoyu Zhan[2]

**1** Key Laboratory of Geological Hazards on Three Gorges Reservoir Area, Ministry of Education, China Three Gorges University, Yichang, China, **2** College of Civil Engineering and Architecture, China Three Gorges University, Yichang, China

\* 202208140011008@ctgu.edu.cn

**Data Availability Statement:** All relevant data are within the manuscript and its Supporting Information files.

**Funding:** This research was funded by the National Natural Science Foundation of China (grant number 51979151); the Natural Science Foundation of

## Abstract

Natural disasters such as landslides often occur on soil slopes in seasonally frozen areas that undergo freeze–thaw cycling. Ecological slope protection is an effective way to prevent such disasters. To explore the change in the mechanical properties of soil under the influence of both root reinforcement and freeze–thaw cycles and its influence on slope stability, the Baijiabao landslide in the Three Gorges Reservoir area was taken as an example. The mechanical properties of soil under different confining pressures, vegetation coverages (VCs) and numbers of freeze–thaw cycles were studied via mechanical tests, such as triaxial compression tests, wave velocity tests and FLAC3D simulations. The results show that the shear strength of a root–soil composite increases with increasing confining pressure and VC and decreases with increasing number of freeze–thaw cycles. Bermuda grass roots and confining pressure jointly improve the durability of soil under freeze–thaw conditions. However, with an increase in the number of freeze–thaw cycles, the resistance of root reinforcement to freeze–thaw action gradually decreases. The observed effect of freeze–thaw cycles on soil degradation was divided into three stages: a significant decrease in strength, a slight decrease in strength and strength stability. Freeze–thaw cycles and VC mainly affect the cohesion of the soil and have little effect on the internal friction angle. Compared with that of a bare soil slope, the safety factor of a slope covered with plants is larger, the maximum displacement of a landslide is smaller, and it is less affected by freezing and thawing. These findings can provide a reference for research on ecological slope protection technology.

## Introduction

A considerable number of small-scale shallow soil landslides have occurred in the Three Gorges Reservoir area in recent years [1–4], posing a significant danger to the safety and stability of reservoir slopes, as well as people's lives and property. A large temperature difference is a major contributor to these landslides [5]. Colder temperatures cause the water in the soil to continuously undergo the liquid–solid phase transition, destroying the bonds between soil

Hubei Province Outstanding Youth Project(Grant number 2021CFA090); and the Three Gorges Key Laboratory of Geological Hazards of the Ministry of Education (China Three Gorges University) (Grant number 2020KDZ07). The funders had no role in study design, data collection and analysis, decision to publish, or preparation of the manuscript.

**Competing interests:** Conflicts of Interest: The authors declare no conflict of interest. Statement: we confirm that all the experiments/ protocols were performed with relevant institutional, national, and international guidelines and legislation

particles and reducing the strength and stability of the soil as a whole. In particular, in the process of frost heave, the formation of ice crystals in the soil increases the distance between the soil particles, resulting in the volume expansion of the soil, which causes damage to and deformation of the soil. Compared with that of ordinary soil, the water content of soil in water level fluctuation areas is much greater, and the performance degradation caused by freeze–thaw cycles is more severe [1, 5]. As a result, steps must be taken to mitigate the impacts of freeze–thaw cycles on slopes, and plant roots play a significant role in maintaining shallow slope stability [6, 7]. Vegetation has been widely utilized in ecological projects in recent years to efficiently reduce soil erosion and stabilize slopes [8, 9].

Freeze–thaw cycles have a great influence on the structural and physical and mechanical properties of soils, and many scholars have conducted series of experimental studies on the effects of freeze–thaw cycles on various soils. The results of Chamberlain et al. [10–12] showed that freezing of the water in soil led to crack formation and increased porosity. The change in the basic physical properties of soil is also related to this phenomenon. Yang et al. [13–17] reported that the shear strength of soil decreased after freeze–thaw cycling. Some scholars have further explored the influence of freeze–thaw cycles on the shear strength parameters of rock and soil [18]. The test results of Aoyama et al. [19] showed the freeze–thaw cycling will lead to the weakening of soil cohesion, although the internal friction angle will not change much. The research results of Ogata et al. [20] showed that the internal friction angle of soil increased as the number of freeze–thaw cycles increased, although the cohesion decreased. Wang et al. [21] measured the shear strength and elastic modulus of soil before and after freezing and thawing and reported that the internal friction angle increased while the elastic modulus and cohesion decreased.

Appropriate reinforcement measures should be adopted to resist the deterioration effect of freeze–thaw cycles on soil. Lime reinforcement, fiber-reinforced concrete reinforcement and other technologies are widely used in the field of soil reinforcement [22]. Numerous studies have shown that while plants improve the quality of the ecological environment, their roots can also be seen as a natural fiber that plays a significant role in reinforcing the shallow soil of the slope, significantly increasing soil strength [23, 24]. Ding et al. [25] used plant roots of three vegetation types to reinforce the soil of high and steep slopes. The results showed that the soil strength of the three vegetation types improved to varying degrees. Noorasyikin et al. [26] used bermudagrass to reinforce sand and clay. Bermudagrass had a greater effect on the clay than on the sand and enhanced the cohesion of the clay. Through pull-out tests, Su et al. [27] quantitatively characterized the anchorage effect of a root system on soil.

Although there has been significant research on the reinforcement effect of roots on soil and the deterioration effect of freeze–thaw cycles on soil, there is limited research on the mechanical properties of soil under the combined action of both processes. Additionally, there are few indicators available for the quantitative evaluation of the interaction between the two. This study explores the influence of root content, confining pressure, and freeze–thaw cycles on the mechanical properties of soil and the resistance mechanism of roots to freeze–thaw cycles. The resistance effects of roots and freeze–thaw cycles are quantitatively characterized, and the stability of the Baijiabao landslide under different working conditions is analyzed via simulation model calculations. This study is based on unconsolidated–undrained (UU) tests, wave velocity tests, and optical microscopic observations.

## Materials and methods

### Test materials and research areas

The plant roots and soil used in this study were bermudagrass roots and undisturbed soil from the Baijiabao landslide in the Three Gorges Reservoir area. The Baijiabao landslide is located

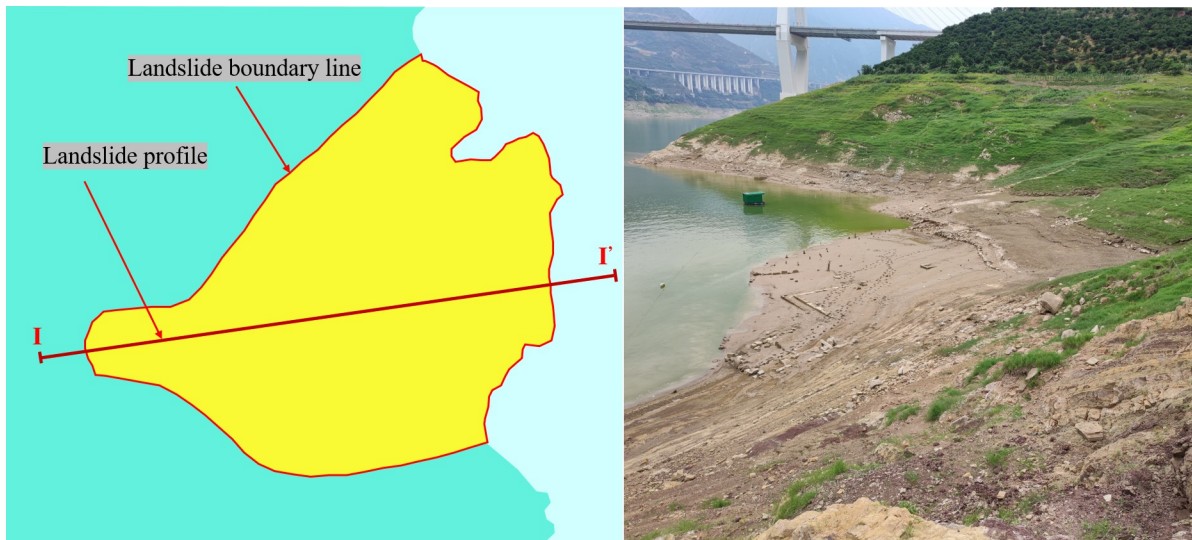

**Fig 1. The morphology of the landslide.**

in Hubei Province, China. Its altitude is 110°45′33.4″, and its latitude is 30°58′59.9′. Overall, the landslide is higher in the west and lower in the east. Homologous gullies split the northern and southern sides of the landslide body. The boundary of the landslide along both sides and the trailing edge is a contact surface between rock and soil. The trailing edge's terrain is shaped like a circular chair, with the front and center portions being gentle and the back portion being steep. The landslide body has a slope of 10° to 30°. The planar form of the landslide body is tongue-shaped, and it is stepped in profile. The landslide body is 550 m long and 400 m wide. The thickest part of the landslide body is approximately 80 m thick, and the average thickness is 45 m. Its volume is approximately $990 \times 10^4$ m³. Since the landslide formed, its subsequent movement has exhibited seasonal variations. Since monitoring began in November 2006, the landslide has experienced periodic deformation and slip in the winter of each year. To decrease the displacement rate, a large amount of bermudagrass was planted on the slope surface to reinforce the soil. The maximum freezing depth of the soil is approximately 30 cm, and the reinforcement depth of bermudagrass roots is 10~30 cm. The morphology of the landslide is shown in Fig 1.

The collected soil samples were subjected to numerous soil tests according to the "Standard for Soil Test Method" (GB/T50123-1999) specification [28]. The physical parameters obtained are shown in Table 1. The soil samples were classified as a clay of high plasticity (CH) according to the Unified Soil Classification System (USCS). The gradation curve of the soil sample is shown in Fig 2.

## Sample preparation

The sample preparation method was similar to that in References [6, 22, 24]. The soil samples were dried by a dryer, and the larger stones and impurities were removed. The prepared

**Table 1. Physical parameters of the soil samples.**

| density $\rho$ (g·cm⁻³) | water content $w$ (%) | maximum dry density $\rho_{dmax}$ (g·cm⁻³) | optimum moisture content $w_{op}$ (%) | soil particle proportion Gs | plasticity index $I_P$ |
|---|---|---|---|---|---|
| 1.977 | 21.54 | 1.9082 | 18 | 2.67 | 18.87 |

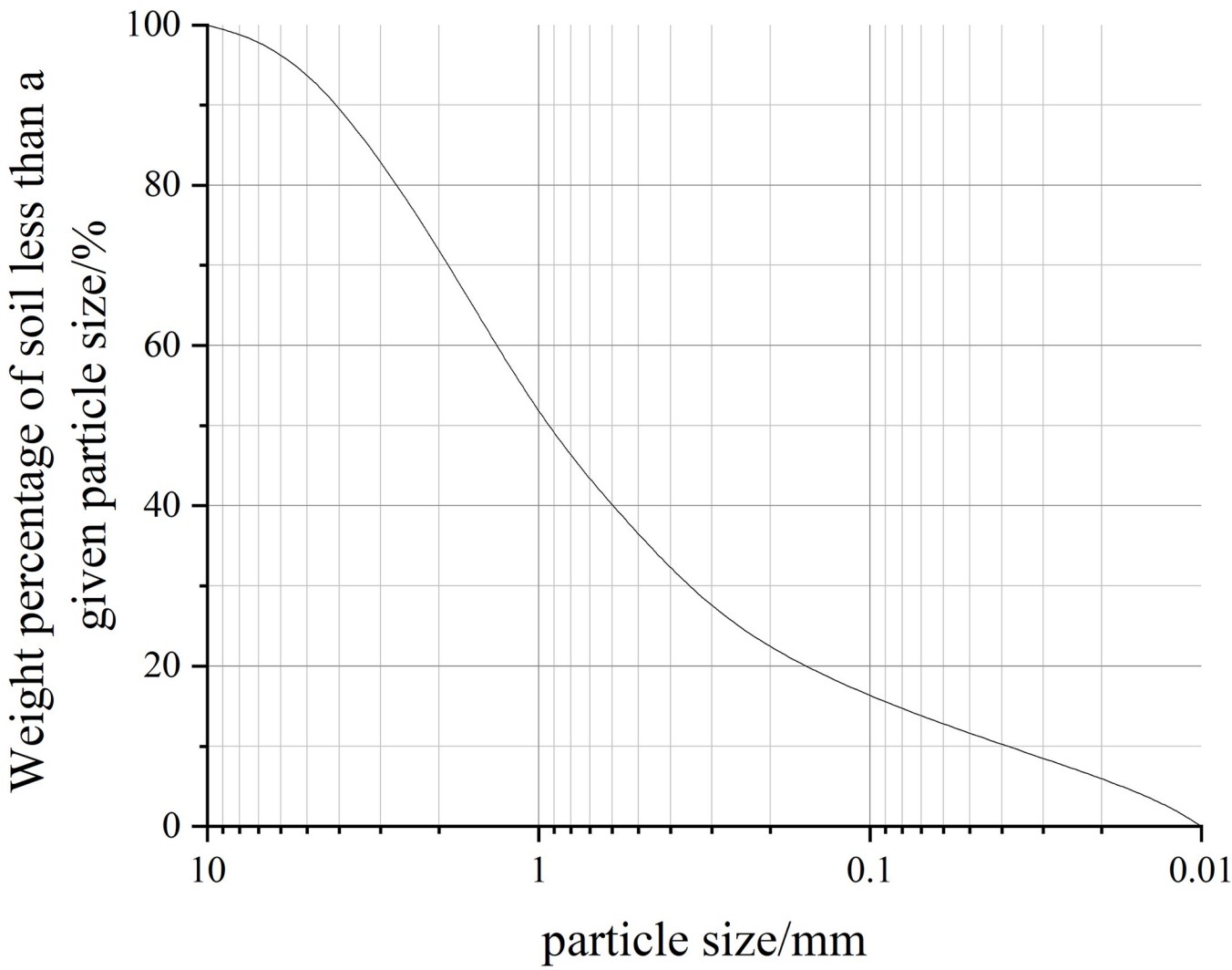

**Fig 2. The gradation curve of the soil sample.**

Bermuda grass roots were washed and allowed to dry naturally. The target dry density was 1.63 g·cm$^3$ to ensure that the compactness of the sample was approximately 85%. The total mass of pure water required to reach the target moisture content was weighed according to the dry soil mass. Pure water was sprayed on the surface of the dry soil with a sprayer and uniformly mixed in with a stirrer until the sample reached the target moisture content of 18%.

The root-containing remolded soil samples were prepared by the compression method [26]. The preparation of root-containing samples by the this method was completed by an automatic pressing prototype with a static constant pressure. The sample preparation process is shown in Fig 3. The mass of each sample was 721.2 g. The configured root–soil mixtures were loaded into containers and mixed evenly. The weighed soil samples were divided into 5 parts for sample preparation. Cylindrical samples with dimensions of 61.8 mm × 125 mm were created in accordance with the specifications of the testing device. Each pressed sample was sealed with cling film and placed in a sealed container to prevent water loss. Five groups of samples with different root contents were prepared. Five different numbers of freeze–thaw

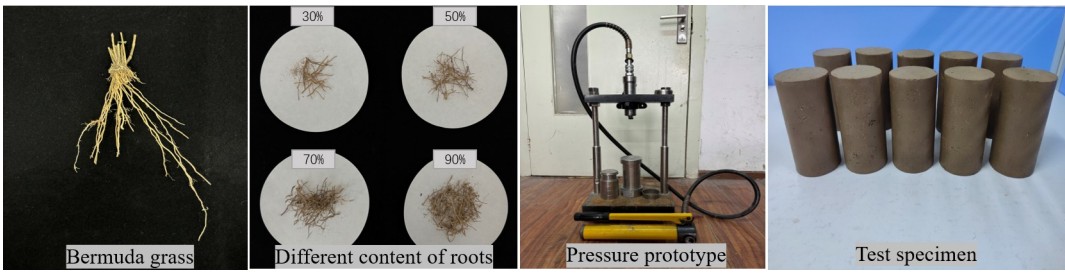

**Fig 3. The Sample preparation process.**

cycles were applied to the five samples in each group with a different root content; 4 samples were tested for each condition, for a total of 100 samples.

## Test scheme

The test instrument used in this paper is a saturated-unsaturated soil stress path triaxial test system jointly developed by Nanjing Soil Instrument Factory and the authors' research group, as shown in Fig 5.

Notably, previous research using traditional methods has often demonstrated a relationship between root content and soil strength [29–31]. However, because the roots of naturally growing plants are buried underground, it is impossible to intuitively judge the reinforcement of soil through the growth state of vegetation, making some research less instructive for actual slope reinforcement projects. To address this issue, this work used the point-frequency (PF) method [32, 33] to evaluate plant cover within a fixed range of slope surfaces. The soil in this range was measured to calculate the root content (root mass/dry soil mass), and a relationship was established between the vegetation coverage (VC) and root content (Fig 4). The root contents were 0%, 0.0856%, 0.1262%, 0.1786%, and 0.2149%, corresponding to VC contents of 0%, 30%, 50%, 70%, and 90%, respectively, so that the reinforcement of slope surface soil can be judged according to the growth state of the vegetation.

D0, D2, D4, D6, and D8 represent root–soil composite samples with 0, 2, 4, 6, and 8 freeze–thaw cycles, respectively, and G0, G3, G5, G7, and G9 represent root–soil composite samples with 0%, 30%, 50%, 70%, and 90% VC, respectively. The test scheme is shown in Table 2.

The Three Gorges Reservoir region is a seasonally frozen soil area, according to temperature data for the Yichang sector of the Three Gorges Reservoir area from 1971 to 2022 published by the China Meteorological Center's data sharing platform. The data suggest that the temperature difference between day and night was significant in the winter, with the minimum temperature at night reaching -9.8˚C and the highest temperature during the day reaching 23˚C. As a result, in this experiment, the melting temperature was set to 25˚C, and the freezing temperature was set to -10˚C: a freeze–thaw cycle of 12 hours of freezing and 12 hours of thawing was employed. The uniform distribution of root aggregates in the sample is obviously different from the actual situation of root growth from the surface downward into the soil in the field; therefore, all surfaces of the samples were regarded as freeze–thaw surfaces, and circumferential freeze–thaw was adopted instead of unidirectional freeze--thaw, which is more reasonable and is consistent with the freeze–thaw method in References [12–14]. According to our preliminary experimental results, the soil strength does not decrease after more than 8 freeze–thaw cycles. Therefore, the numbers of freeze–thaw cycles investigated here are 0, 2, 4, 6, and 8. The Test steps is shown in Fig 5.

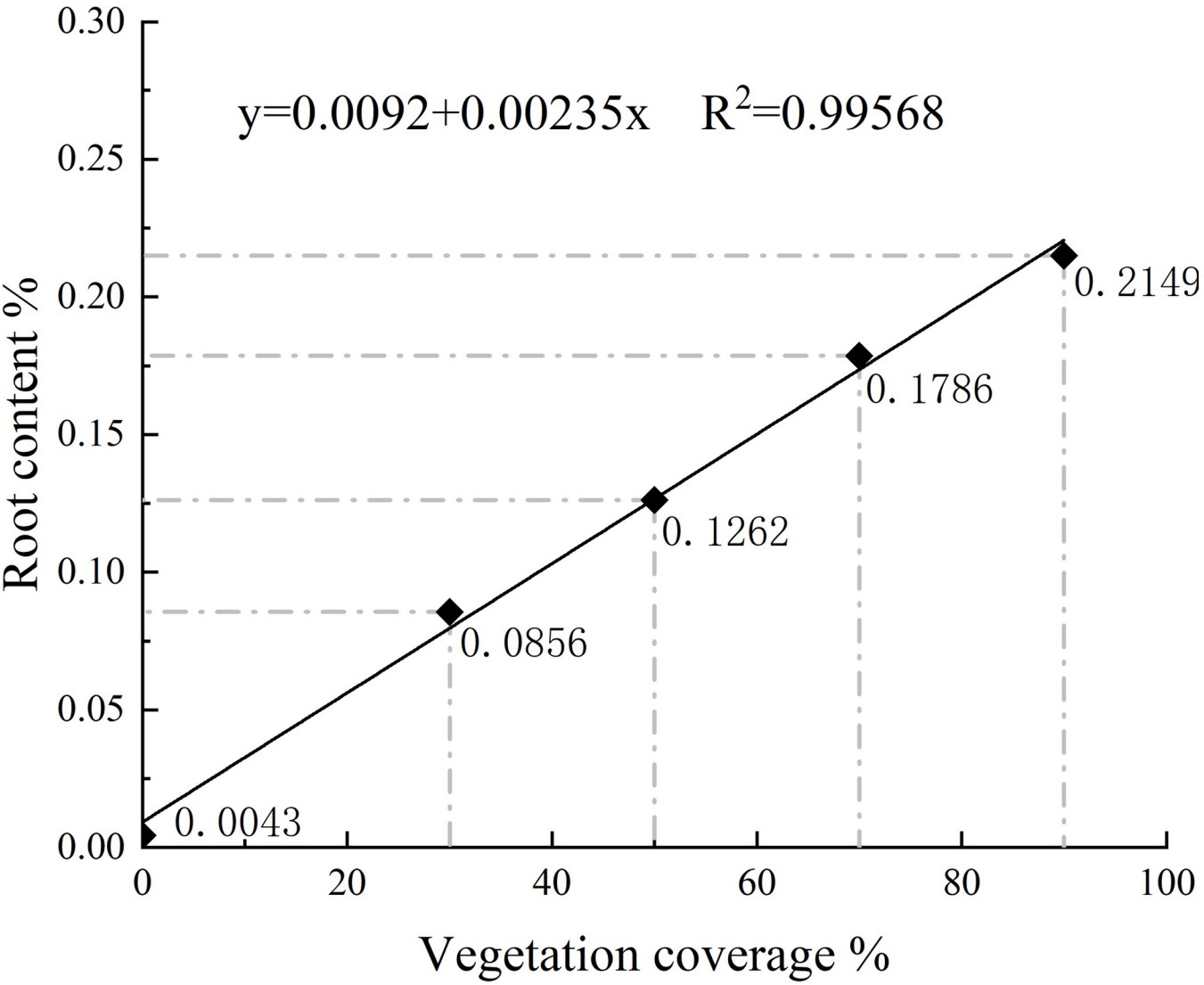

**Fig 4. The vegetation coverage (VC) and root content.**

After freeze–thaw treatment, a sample was subjected to a wave velocity test with a YL-SWT shear wave velocity tester. During the test, a sensor and transmitter were installed on the soil sample to ensure that the position was accurate and stable. Then, the data acquisition device and the recorder were connected, the sound wave signal was transmitted, the sound wave propagation time was recorded, and the wave velocity was calculated. Subsequently, UU tests [34] were conducted under confining pressures of 50 kPa, 100 kPa, and 200 kPa. The axial strain rate range for the UU tests, as per the geotechnical test standard, was set to 1%~3%, which is equivalent to a loading rate of 1.25~3.75 mm·min-1. During the tests, the loading rate was set to 1.25 mm·min-1 based on the sample size. A test was considered complete when the stress–strain curve reached its peak shear strength value and the axial strain continued to increase by approximately 3% to 5% beyond the peak value. If there was no peak, the test ended when the axial strain reached 15%. Fig 5 illustrates the detailed test steps.

**Table 2. Test scheme.**

| group | water content $w$ (%) | dry density $\rho_d$ (g·cm$^{-3}$) | VC (%) | freeze–thaw cycles | confining pressure $\sigma_2 = \sigma_3$ (kPa) |
|---|---|---|---|---|---|
| 1 | 18.0 | 1.63 | 0 | 0, 2, 4, 6, and 8 | 50 |
| | | | | | 100 |
| | | | | | 200 |
| 2 | | | 30 | 0, 2, 4, 6, and 8 | 50 |
| | | | | | 100 |
| | | | | | 200 |
| 3 | | | 50 | 0, 2, 4, 6, and 8 | 50 |
| | | | | | 100 |
| | | | | | 200 |
| 4 | | | 70 | 0, 2, 4, 6, and 8 | 50 |
| | | | | | 100 |
| | | | | | 200 |
| 5 | | | 90 | 0, 2, 4, 6, and 8 | 50 |
| | | | | | 100 |
| | | | | | 200 |

## Results and dissection

### Stress–strain curve characteristics

The stress–strain curves of the samples tested with different VCs and confining pressures and after different numbers of freeze–thaw cycles are shown in Fig 6. There was no noticeable peak stress in either the bare soil or root-containing soil, and there was no softening phenomenon in the corresponding stress–strain curves. The deviatoric stress increased as the axial strain increased, and the rate of increase in stress increased more quickly in the early stage of testing

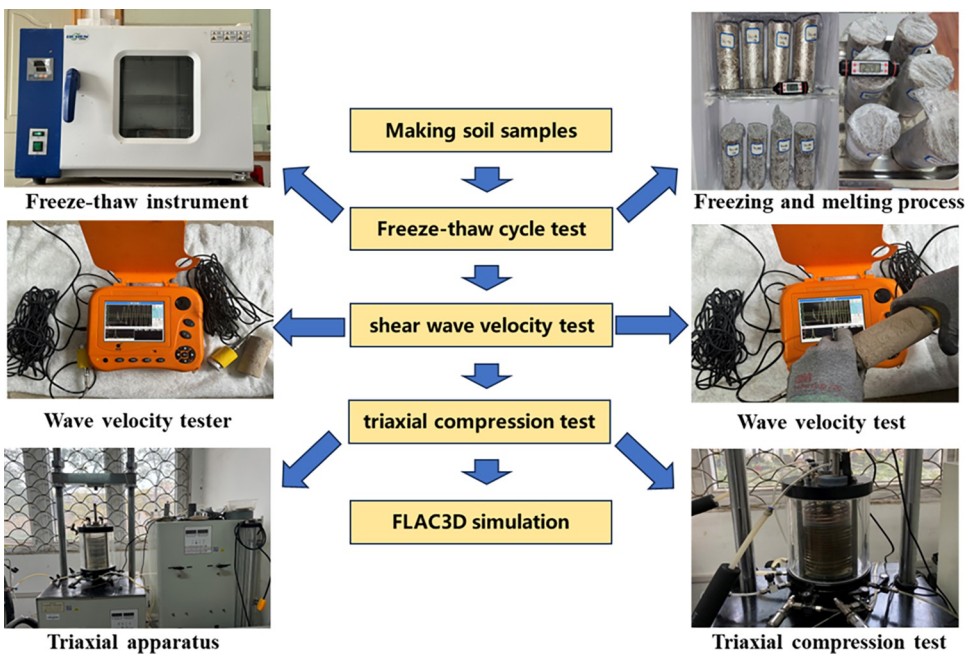

**Fig 5. The test steps.**

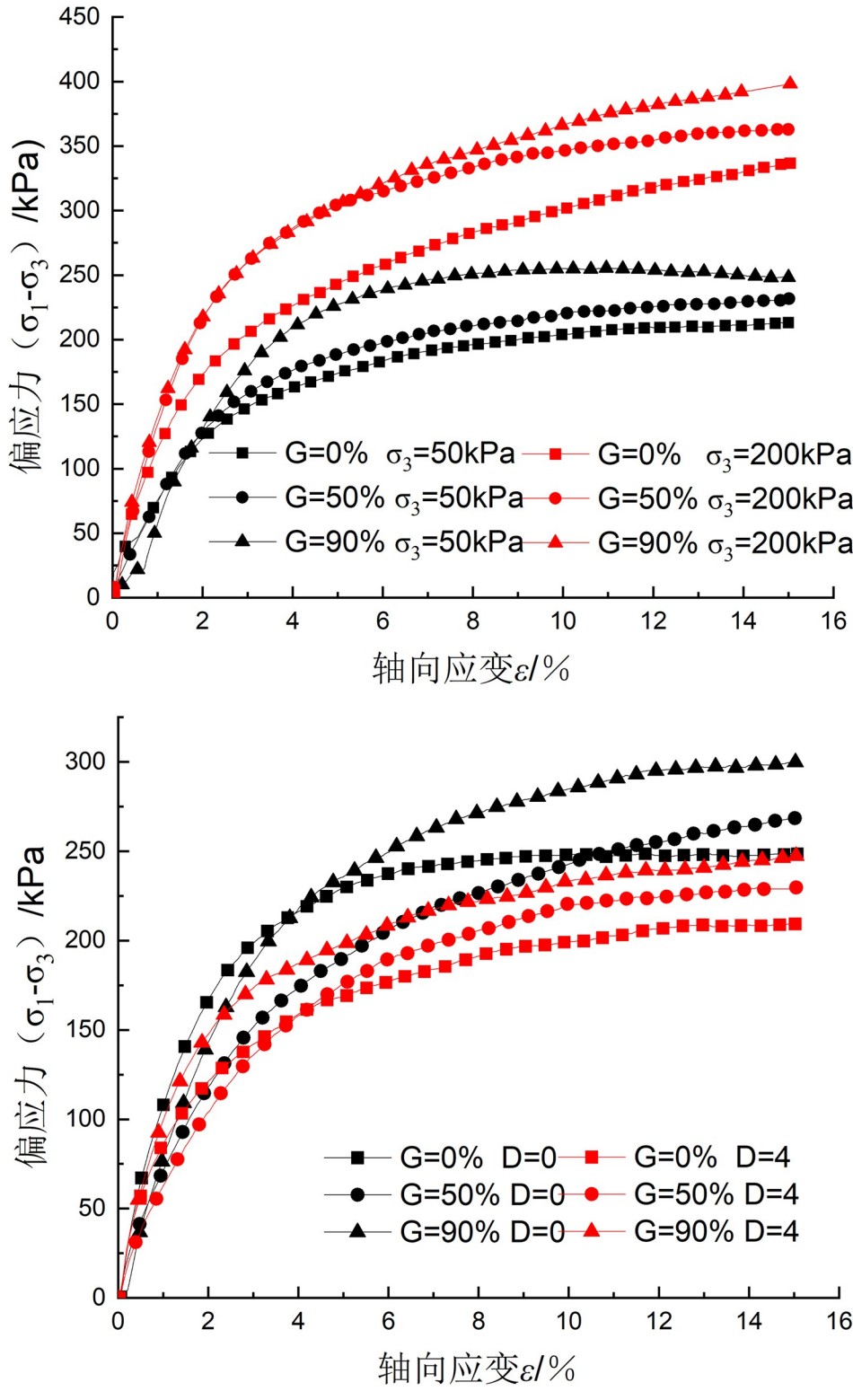

**Fig 6.** Sample stress–strain curves: (a) The influence of VC and confining pressure; (b) The influence of VC and number of freeze–thaw cycles.

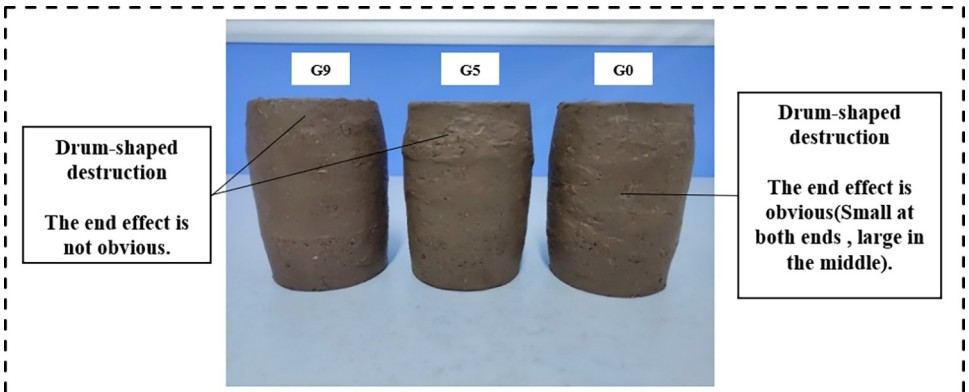

**Fig 7. Shear failure image after 8 freeze–thaw cycles.**

and then slowed and eventually stabilized in the late stage. The continuous strain hardening phenomenon becomes increasingly clear as the confining pressure and VC increase, as shown in Fig 6A. Fig 6B indicates that under constant confining pressure and VC conditions, the deviatoric stress decreases as the number of freeze–thaw cycles increases, showing that the freeze–thaw cycling weakens the soil.

Images of the soil samples after 8 freeze–thaw cycles are shown in Fig 7 for morphological comparison. Many folds and holes formed on the surface of the three samples, and few soil particles detached, suggesting that the freeze–thaw activity disrupted the internal structure of the soil. The sample morphology demonstrated a typical end effect (wide in the middle, narrow at both ends), and the severity of the end effect decreased as VC increased, indicating that the root system could effectively inhibit the compaction and expansion caused by loading and frost heave.

## Shear strength characteristics

Fig 8A shows the effect of VC on the shear strength of the root–soil composites under a 200 kPa confining pressure and after various numbers of freeze–thaw cycles. In the absence of freeze–thaw action, the roots increased the soil strength by 18.10%, and the reinforcement effect was remarkable. The slope of the fitting line gradually decreased as the number of freeze–thaw cycles increased; the slope decreased from 0.661 to 0.338, showing that the freeze–thaw treatment diminished the reinforcing effect of the roots on the soil. These findings are in good agreement with those of previous studies [14, 17]. When the number of freeze–thaw cycles was 8, the roots only increased the soil strength by 13.69%.

The effect of the number of freeze–thaw cycles on the shear strength of the root–soil composites with various VC contents under a confining pressure of 200 kPa is reflected in the data shown in Fig 8B. It is evident that the correlation between the shear strength and the number of freeze–thaw cycles is exponentially negative. The freeze–thaw treatment reduces the soil shear strength more significantly in the early stage of testing than in the late stage, which was also reported in previous studies [15, 19].

To quantitatively reveal the resistance of roots to freeze–thaw deterioration, the freeze–thaw–root combined action factor $\lambda_{i-j}$ was defined.

$$\lambda_{i-j} = \frac{\tau_{i-0} - \tau_{0-0}}{\tau_{i-0} - \tau_{i-j}} \tag{1}$$

where $i$ is the VC and $j$ is the number of freeze–thaw cycles.

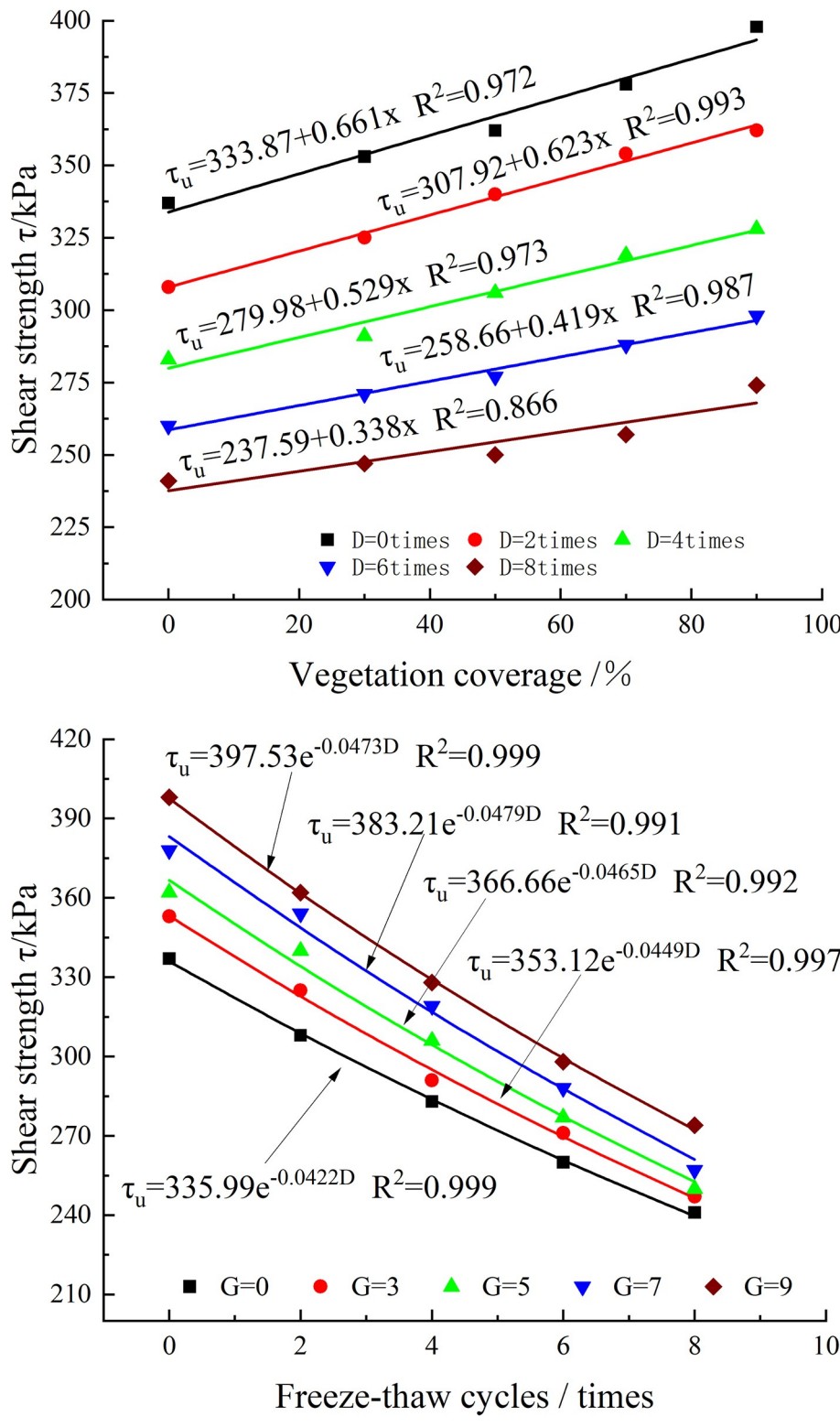

**Fig 8. Shear strength variation curve.** (a) Relationship between shear strength and VC ($\sigma_3$ = 200 kPa); (b) Relationship between shear strength and number of freeze–thaw cycles ($\sigma_3$ = 200 kPa).

**Table 3. The $\lambda_{i-j}$ values for different VC and freeze–thaw cycle test conditions.**

| $\lambda_{i-j}$ | i = 3 | i = 5 | i = 7 | i = 9 |
|---|---|---|---|---|
| j = 2 | 0.57 | 1.14 | 1.71 | 1.69 |
| j = 4 | 0.26 | 0.45 | 0.69 | 0.87 |
| j = 6 | 0.20 | 0.29 | 0.46 | 0.61 |
| j = 8 | 0.15 | 0.22 | 0.34 | 0.49 |

According to $\lambda_{i-j}$, the extent to which the VC reinforcement effect of the soil can offset the freeze–thaw deterioration effect of the soil can be intuitively determined. When $\lambda_{i-j}$ is greater than 1, the reinforcement effect is greater than the deterioration effect. When $\lambda_{i-j}$ is less than 1, $\lambda_{i-j}$ is the proportion of the reinforcement effect offsetting the freeze–thaw effect. According to the results of $\lambda_{i-j}$ calculated from Fig 8, the data in Table 3 were obtained.

According to Table 3, $\lambda$ is greater than 1 only when $j = 2$ and $i = 5$, 7, and 9, which means that only the reinforcement of roots in these three cases can completely offset the degradation of freeze–thaw cycles. When $i \geq 4$, no matter the value of $j$, the degradation of soil strength caused by freeze–thaw cycles cannot be completely offset. When $i = 8$, the maximum offset that the reinforcement effect of the root system can provide is less than half the deterioration effect (0.49). In addition, under the condition that $i$ is constant, although $\lambda$ decreases as $j$ increases, the rate of decrease gradually slows until it tends to stabilize. For example, the difference between $\lambda_{3-2}$ and $\lambda_{3-4}$ is 0.31, the difference between $\lambda_{3-4}$ and $\lambda_{3-6}$ is 0.06, and the difference between $\lambda_{3-6}$ and $\lambda_{3-8}$ is 0.05. This shows that the rate of reduction in soil strength will gradually decrease with increasing number of freeze–thaw cycles until it tends to stabilize, i.e., the degradation of soil by freeze–thaw cycles will inevitably progress through three stages: a large strength deterioration stage, a small strength reduction stage and a stable strength stage.

## Shear strength parameters

By studying the variation in the shear strength parameters (c, φ) of soil samples with VC after freeze–thaw action, the influence of the two on the basic physical and mechanical properties of soil can be analyzed. Therefore, we draw the Mohr stress circle and calculate the c and φ values of the soil samples according to the triaxial tests under three confining pressures of 50 kPa, 100 kPa and 200 kPa. A plane rectangular coordinate system is constructed, and the changes in c and φ of the different soil samples are shown in Fig 9A and 9B. In Fig 9A, with increasing VC, the cohesion shows different degrees of increase. When the number of freeze–thaw cycles is 0, 2, 4, 6 and 8, the cohesion of the root–soil composites with VC contents of 30%, 50%, 70% and 90% increase by 2.89%~13.04%, 9.39%~22.56%, 14.39%~20.53%, 9.55%~30.6% and 20.06% ~73.25%, respectively, compared with that of the rootless soil. In contrast, the internal friction angle does not change significantly, and the overall change range is small.

To quantitatively reveal the influence of the two on c and φ, the cohesive damage residual coefficient $K_c$ and the internal friction angle damage residual coefficient $K_\varphi$ are defined.

$$K_c = \frac{c_n}{c_0} \tag{2}$$

$$K_\varphi = \frac{\varphi_n}{\varphi_0} \tag{3}$$

where $c_n$ and $c_0$ are the cohesion values after n freeze–thaw cycles and zero freeze–thaw cycles,

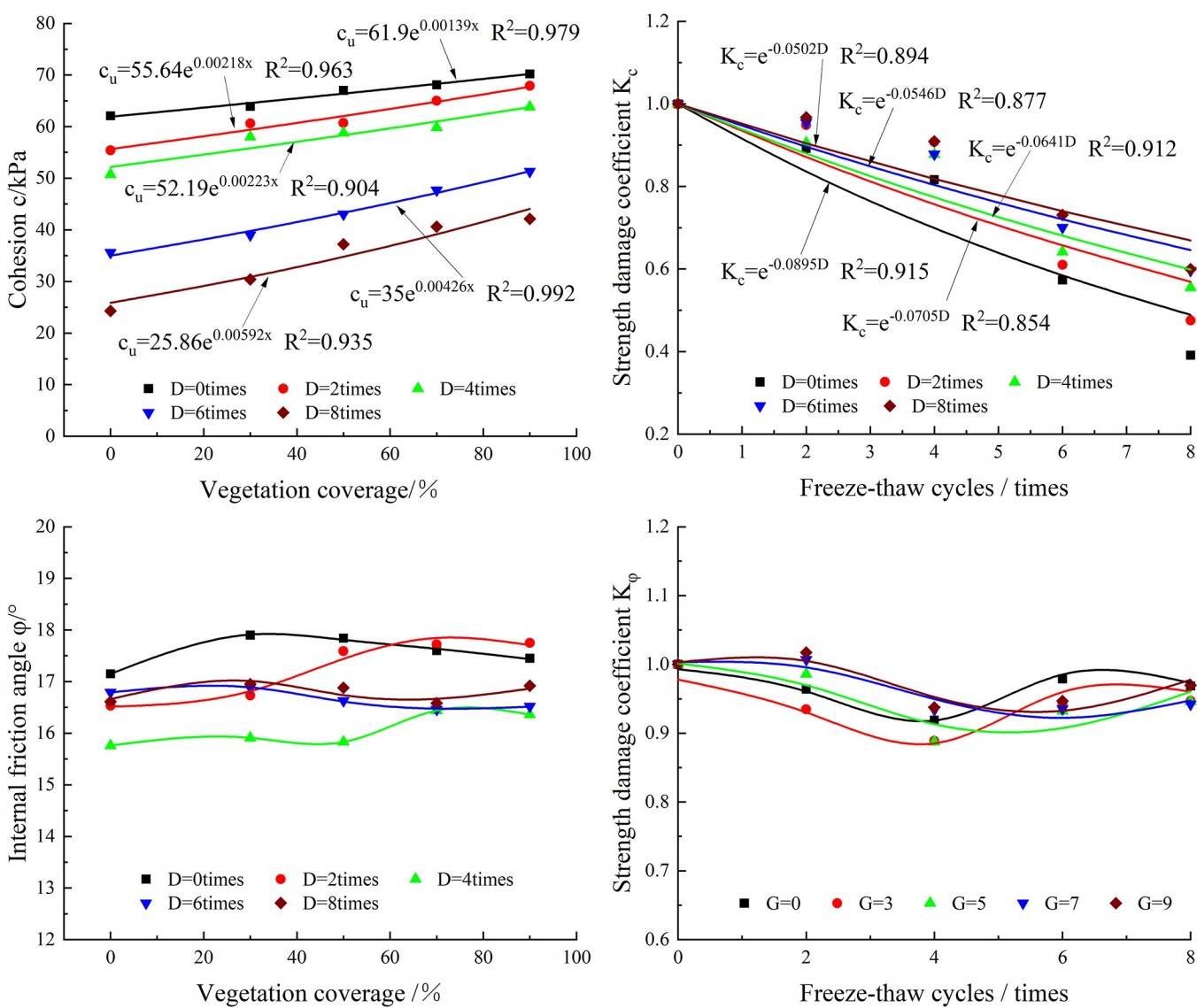

**Fig 9. The relationship of the shear strength parameter and strength damage coefficient with the number of freeze–thaw cycles.** (a) cohesion; (b) internal friction angle; (c) Kc; and (d) K$_\varphi$.

respectively, and $\varphi_n$ and $\varphi_0$ are the internal friction angle values after n freeze–thaw cycles and zero freeze–thaw cycles, respectively.

The variations in $K_C$ and $K_\varphi$ are shown in Fig 9C and 9D, respectively.

## Shear wave velocity

The wave velocity test results are shown in Fig 10. A decrease in the wave velocity is a sign of damage and destruction of the internal structure of the soil by freeze–thaw action. The lower the wave velocity, the more internal pores there are in the soil. The figure shows that the decrease in the wave velocity for all the samples is greatest after the first two freeze–thaw cycles; thereafter, it steadily decreases and begins to stabilize, which is consistent with the change in the freeze–thaw–root combined action factor $\lambda_{i-j}$. Additionally, as VC increases due to an

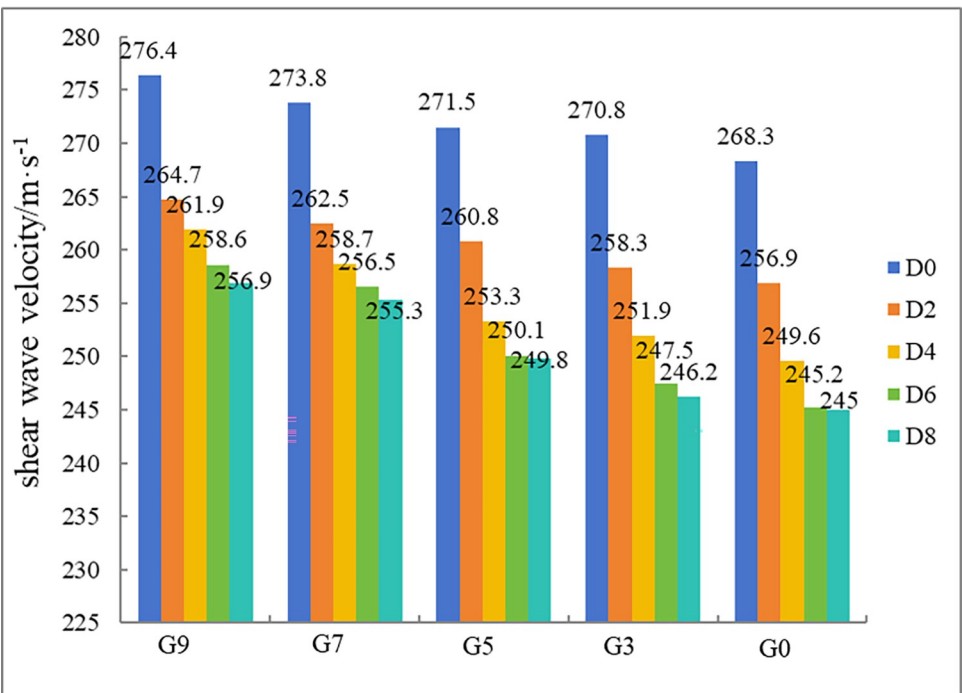

**Fig 10. Shear wave velocity test results.**

increase in the fraction of root material in the sample, the initial wave velocity of the sample decreases.

## Changes in the soil microstructure

Fig 11 shows the microstructure of the bare soil and root–soil composites before and after freezing and thawing under a microscope. Prior to freezing and thawing, the soil particles in both the root–soil composite and bare soil samples were intact. However, the freezing and thawing process altered the pore distribution properties, resulting in an increased porosity and pore size of the samples. After eight cycles of freezing and thawing, both types of samples exhibited numerous holes, some of which developed into microfissures. Before freezing and thawing, the roots inside the root–soil composite were intertwined to form a complex supporting system, and the roots and soil were in close contact at root–soil interfaces. However, after the freezing and thawing process, holes appeared around the root system. This was due to the presence of water within the roots. During frost heave, water froze around the root, causing it to separate from the soil interface. Upon thawing, microholes formed around the root system.

## Slope stability simulation

FLAC3D was used to simulate the deformation of the shallow soil of the Baijiabao landslide body, and the safety factor was calculated to explore the influence of VC and freeze–thaw cycles on slope stability. The calculation model selects the I-I' section of the Baijiabao landslide as the calculation section and establishes a two-dimensional model, as shown in Fig 12. The length, width and height of the model are 800 m, 100 m and 320 m, respectively, which are the same as the actual landslide size. Fixed constraints are set on the bottom, front and rear boundaries of the model, normal constraints are set on the side boundaries, and the top surface is a free boundary. The mesh of the Baijiabao landslide model is divided into triangular node

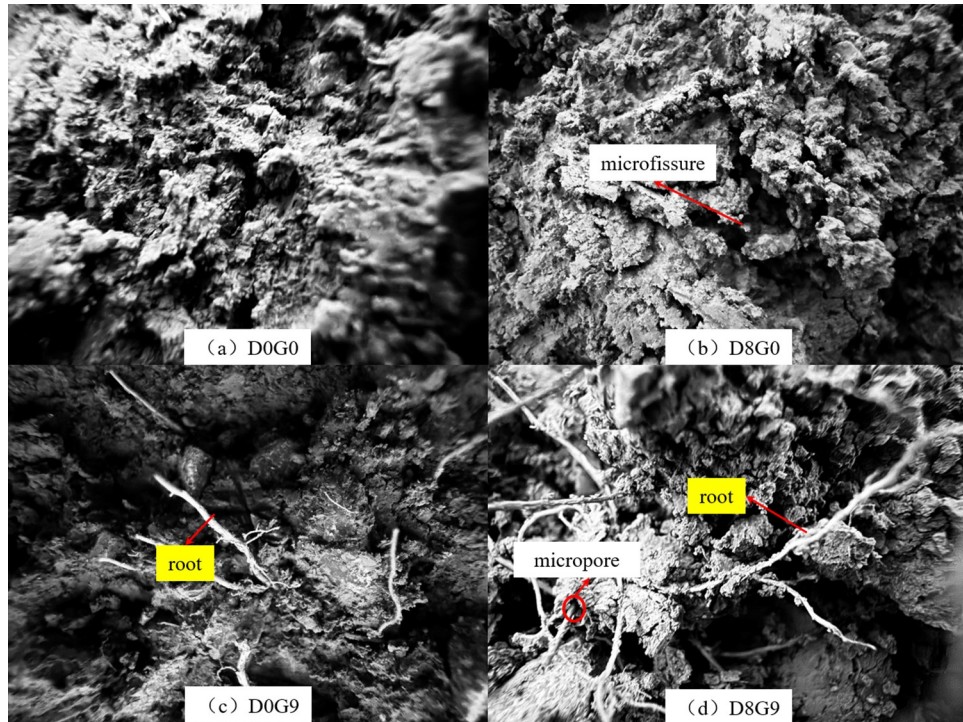

**Fig 11. Microstructure of the bare soil and root–soil composite samples before and after freezing and thawing.**

elements. A total of 29945 elements were generated, covering a total of 6484 nodes. The model follows the Mohr–Coulomb yield criterion.

Many geological survey reports on the Baijiabao landslide were consulted, as were the physical and mechanical parameter data of shallow rock and soil masses of the same type of landslide in the Three Gorges Reservoir area. After comprehensive consideration, the simulation calculation parameters of the landslide deformation displacement were determined, as shown in Table 4.

In this work, the strength reduction method is used to reduce the c and φ values of the bare soil and root–soil composite, and the shear strength parameter of the reduced soil is substituted into the model for calculation. Finally, the displacement and safety factor of the bare soil slope and vegetation-covered slope are obtained.

## Slope stability without freeze–thaw cycles

Table 5 shows that as the VC increases, the maximum displacement of the slope decreases, while the safety factor increases. The soil displacement of the simulated vegetation-covered slope is 2.4%~4.1% less than that of the bare soil slope. When VC is 90%, the slope safety factor reaches 2.43, which is 6.11% greater than that of the bare soil slope.

## Slope stability after freeze–thaw cycles

To explore the influence of freeze–thaw cycling on the stability of the shallow soil of the Baijiabao landslide, the maximum displacement and safety factor of the bare soil slope and VC 90% slope after freeze–thaw cycling were compared. The results are shown in Table 6.

The change trends of the safety factor and displacement of the bare soil slope and vegetation-covered slope are roughly the same. In the process of enhancing the freeze–thaw effect of

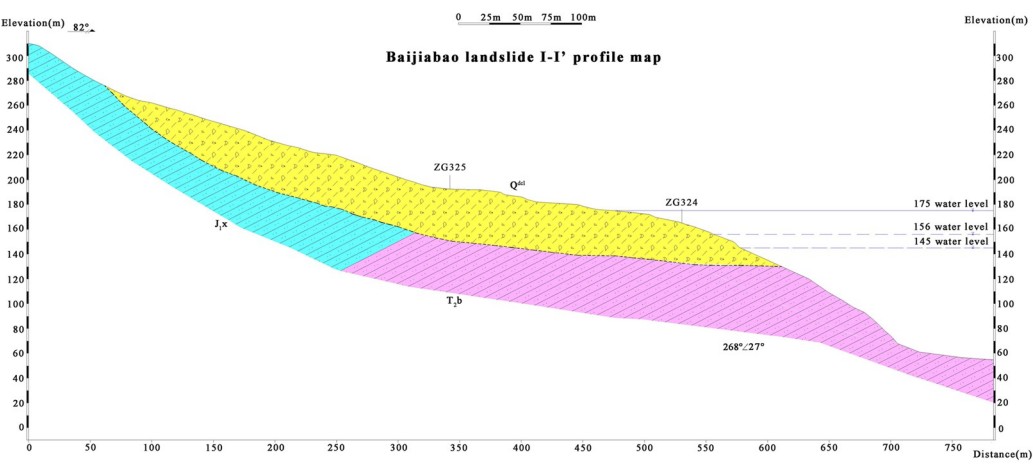

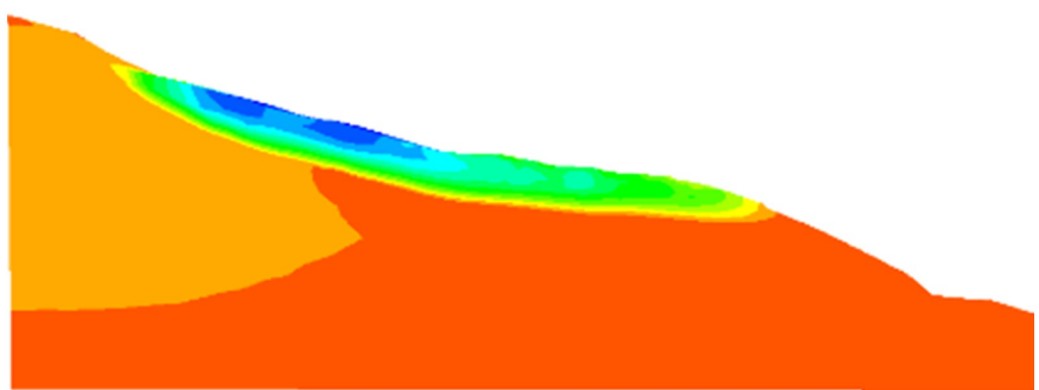

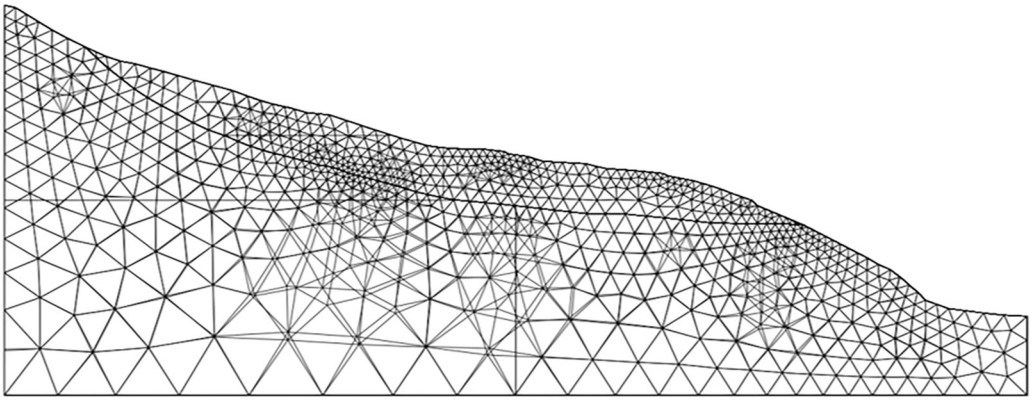

**Fig 12. Baijiabao landslide simulation model.** (a) I-I' profile (Fig 1); (b) Numerical simulation effect diagram; (c) Grids and nodes.

the shallow surface soil of the slope, the deformation of the shallow surface soil of the slope increases, and the safety factor decreases gradually, which fully shows that the freeze–thaw cycling will have an adverse effect on the stability of the shallow surface soil of the slope. When the bare soil slope experienced 8 freeze–thaw cycles, the displacement of the soil was the largest, increasing by 8.78% compared with that without freeze–thaw cycles, while the displacement of the vegetation-covered slope increased by only 5.21%. Under the same number of

**Table 4. Deformation calculation parameters of the Baijiabao landslide.**

| Parameter name | Density $P$ (kg/m³) | Shear modulus $G$ (Pa) | Bulk modulus $K$ (Pa) | Poisson's ratio $u$ | Bonding force $C$ (Pa) | Internal friction angle $\Phi$ (°) |
|---|---|---|---|---|---|---|
| Bedrock | 2800 | 1.08e10 | 1.56e10 | 0.22 | 2.45e6 | 42 |
| Sliding mass | 2190 | 5.25e7 | 7.45e7 | 0.25 | 1.41e4 | 26 |
| Sliding belt | 2090 | 3.5e7 | 4e6 | 0.44 | 6e3 | 22 |

freeze–thaw cycles, the safety factor of the vegetation-covered slope was 2.42%~5.06% smaller than that of the bare soil slope, indicating that the presence of plant roots could effectively inhibit the slip deformation of the shallow soil of the slope. The degradation of the vegetation-covered slope after freeze–thaw action was generally weaker than that of the bare soil slope.

## The influence mechanism of freeze–thaw cycles and roots on the mechanical properties of soil

According to the above test results, combined with the research of other scholars, the attenuation mechanism of the root–soil composite strength due to freeze–thaw action is briefly summarized below (Fig 13):

1. Root reinforcement without freeze–thaw action: Without freeze–thaw action, a root–soil composite, in which the roots are intertwined, forms a uniform and effective "network" support system of roots [35–37]. At the same time, there is strong cohesion between soil particles and between roots and soil particles. At this point, if the root–soil composite experiences shear failure, the aforementioned two bonding pressures must be resisted simultaneously. The greater the number of roots is, the more complicated the support system, the greater the cohesion between the roots and soil particles, and the greater the shear strength of the whole root–soil composite.

2. The decrease in cohesion during freezing and the resistance of roots and high confining pressure: The free water and weakly bound water in soil freeze and expand throughout the freezing process at low temperatures. When the water content of the root system increases,

**Table 5. Slope displacement and safety factor statistics of different VCs.**

| VC (%) | Slope maximum displacement (mm) | Safety factor of slope | Stability improvement ratio |
|---|---|---|---|
| 0 | 11.98 | 2.29 | 0 |
| 30 | 11.69 | 2.38 | 3.93% |
| 50 | 11.56 | 2.39 | 4.37% |
| 70 | 11.56 | 2.41 | 5.24% |
| 90 | 11.49 | 2.43 | 6.11% |

**Table 6. Analysis table of the displacement and safety factor of bare soil and vegetation-covered slopes after different numbers of freeze–thaw cycles.**

| Freeze–thaw cycles | Maximum displacement of slope (mm) | | Safety factor of slope | | The stability reduction rate | |
|---|---|---|---|---|---|---|
| | Bare soil slope | Vegetation slope | Bare soil slope | Vegetation slope | Bare soil slope | Vegetation slope |
| 0 | 11.98 | 11.56 | 2.29 | 2.41 | 0 | 0 |
| 2 | 12.52 | 11.94 | 2.13 | 2.30 | 6.98% | 4.56% |
| 4 | 13.04 | 12.55 | 2.01 | 2.13 | 12.23% | 11.60% |
| 6 | 13.78 | 13.24 | 1.86 | 1.96 | 18.78% | 18.67% |
| 8 | 14.99 | 13.60 | 1.68 | 1.89 | 26.64% | 21.58% |

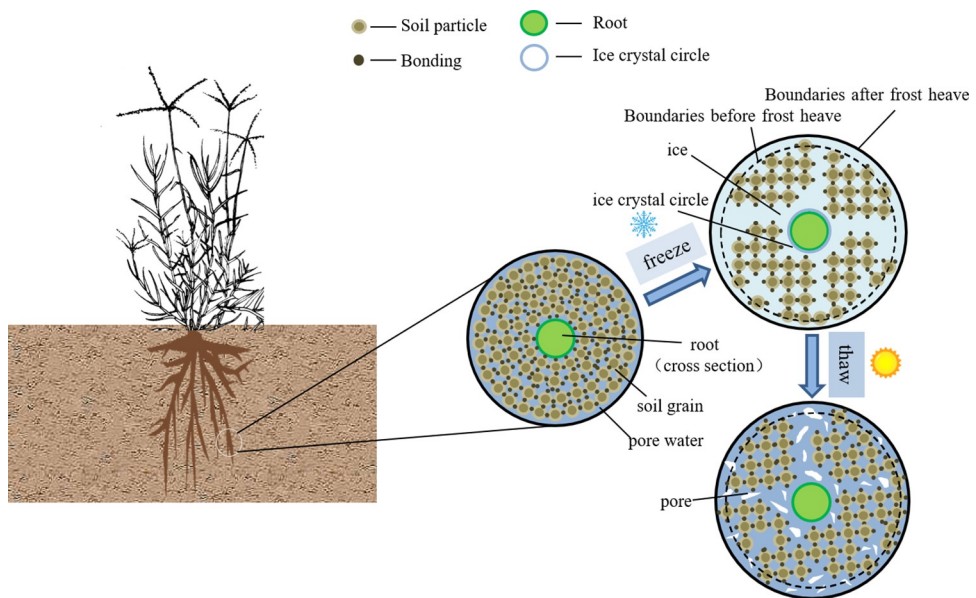

**Fig 13. Strength attenuation mechanism of the root–soil composite under freeze–thaw action.**

an ice crystal ring will form at water surface. The root and soil interfaces with water change into root system-ice crystal bonds and ice crystal-soil bonds. Moreover, the expansion and extrusion of ice crystals breakdown certain soil particles, increasing the porosity of the soil. The interfaces of many soil particles change into ice crystal-soil bonds, resulting in a considerable loss of cohesion [30, 31]. However, the support system established by complex root intertwining remains, limiting the freezing expansion effect. Moreover, a greater external confining pressure suppresses the freezing effect to some extent, preventing the bonding effect from decreasing during freezing.

3. Particles lose contact during melting: As melting begins, ice crystals melt into water, and the root system-ice crystal and ice crystal-soil bonds vanish, resulting in the reduction of particle contact.

4. Repeated freezing and thawing leads to a further reduction in soil particle bonding until the soil becomes stable: When the soil undergoes freeze–thaw action again, the free water and some weakly bound water in the soil will migrate and accumulate again, and its volume will expand again as it freezes. However, the increase in soil pores caused by the phase change expansion of water during the previous freezing effect provides additional space. The general structure of the soil is less damaged by the second freezing than it is by the first freezing. The rate of attenuation slows, but the strength of the soil continues to decrease. The strength of the soil constantly declines as the number of freeze–thaw cycles increases. However, after a given number of freeze–thaw cycles, the impact of the water phase change on the soil structure diminishes, and the strength of the soil tends to stabilize. Due to the strengthening of soil due to the root system, the root–soil composite maintains its strength more than the bare soil does.

## Conclusions

In this paper, freeze–thaw cycle tests and UU tests were carried out on root–soil composites in the Three Gorges Reservoir area, and the effects of freeze–thaw cycles, VC and confining pressure on the mechanical properties of the resulting root–soil composites were analyzed. The

freeze–thaw–root combined action factor $\lambda_{i-j}$ was defined. According to $\lambda_{i-j}$, the reinforcement effect of different root contents on soil can be used to quantitatively express how many freeze–thaw cycles can be resisted. According to the $c$ and $\varphi$ values obtained from the UU tests, a numerical simulation analysis was carried out on the stability of the Baijiabao landslide, and the following conclusions were obtained:

1. With increasing confining pressure and VC, the strain hardening phenomenon of the root–soil composite became more obvious, the end effect was weakened, and the maximum increase in shear strength reached 18.1%, indicating that the Bermuda grass roots and confining pressure together enhance the freeze–thaw durability of the soil.

2. As the number of freeze–thaw cycles increased, the shear strength of the bare soil and root–soil composite decreased in stages. The freeze–thaw deterioration was divided into three stages, namely, the large strength reduction stage, the small strength reduction stage and the stable strength stage.

3. When the number of freeze–thaw cycles was at least 4, $\lambda_{i-j}$ was always less than 1; the degradation of the soil strength caused by the freeze–thaw action could not be completely offset by the root system, regardless of the root content. When the number of freeze–thaw cycles reached 8, the root system could offset less than half the degradation of the soil strength due to freezing and thawing ($\lambda_{i-j} = 0.49$).

4. The cohesive damage residual coefficient $K_c$ increased with increasing VC and decreased with increasing number of freeze–thaw cycles, which is basically consistent with the trend of the change in shear strength. The internal friction angle damage residual coefficient $K_{\varphi}$ always fluctuated around 0.1 and had no obvious correlation with VC or the number of freeze–thaw cycles. This shows that both VC and freeze–thaw action change the strength of soil by affecting the cohesion of the soil rather than the internal friction angle.

5. When there is no freezing or thawing, the maximum displacement of the shallow surface soil of the slope will decrease with increasing VC, which reflects that the plant root system effectively constrains the sliding displacement of the shallow surface of the slope. When experiencing freeze–thaw cycles, the displacement of the shallow surface soil of the landslide increases, and the safety factor decreases. The freeze–thaw effect has a negative effect on the stability of the shallow surface soil of the slope. Under the same number of freeze–thaw cycles, the safety factor of the vegetation-covered slope is 2.42%~5.06% smaller than that of the bare soil slope, indicating that the existence of roots resists the adverse effects of freeze–thaw action on slope stability.

Notably,, in theory, the reinforcement effect of root content on soil becomes limited after a threshold root content. When it exceeds this threshold root content, too many roots will agglomerate in the soil to form a weak surface and destroy the integrity of the soil. However, it is difficult for the coverage of naturally growing Bermuda grass on slopes to exceed 90%; that is, it is difficult for the root content to exceed the maximum root content of 0.22% tested in this paper. Therefore, combined with the actual natural background of the Three Gorges Reservoir area, when using bermudagrass for slope protection, the planting density should be increased as much as possible to increase its reinforcement effect on the soil.

## Supporting information

**S1 File. Original data.**
(ZIP)

## Acknowledgments

The authors gratefully acknowledge the Yangtze River Scientific Research Institute for its help in the field sampling process.

## Author Contributions

**Data curation:** Jingru Jia, Xiaoyu Zhan.

**Investigation:** Zexin Jing, Shun Bao.

**Methodology:** Ruihong Wang, Hao Luo.

**Writing – original draft:** Jingru Jia, Xiaoyu Zhan.

**Writing – review & editing:** Hao Luo.

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
