## [Decision Letter · Decision Letter 0]

13 Feb 2024

PONE-D-23-42173Influence of freeze-thaw and root combined action on soil mechanical characteristics and stability in the water-level-fluctuating zone of Baijiabao landslide in the Three Gorges Reservoir AreaPLOS ONE

Dear Dr. Luo,

Thank you for submitting your manuscript to PLOS ONE. After careful consideration, we feel that it has merit but does not fully meet PLOS ONE’s publication criteria as it currently stands. Therefore, we invite you to submit a revised version of the manuscript that addresses the points raised during the review process.

Dear Authors,

The evaluations from the peer reviewers regarding your submitted work have been duly received. Upon reviewing their feedback, it is evident that they recommend that you revise your manuscript. Therefore, the authors should consider each comment and decide on the best course of action for their research.

We look forward to receiving your revised manuscript.

Kind regards,

Shaker Qaidi

Academic Editor

PLOS ONE

“This research was funded by the National Natural Science Foundation of China (grant number 51979151); the Natural Science Foundation of Hubei Province Outstanding Youth Project(Grant number 2021CFA090); and the Three Gorges Key Laboratory of Geological Hazards of the Ministry of Education (China Three Gorges University) (Grant number 2020KDZ07).”

6. We note that Figures 3, 5 and 13 in your submission contain copyrighted images. All PLOS content is published under the Creative Commons Attribution License (CC BY 4.0), which means that the manuscript, images, and Supporting Information files will be freely available online, and any third party is permitted to access, download, copy, distribute, and use these materials in any way, even commercially, with proper attribution. For more information, see our copyright guidelines: http://journals.plos.org/plosone/s/licenses-and-copyright.

1. You may seek permission from the original copyright holder of Figures 3, 5 and 13 to publish the content specifically under the CC BY 4.0 license.

Reviewers' comments:

Reviewer's Responses to Questions

**Comments to the Author**

1. Is the manuscript technically sound, and do the data support the conclusions?

Reviewer #1: Yes

Reviewer #2: Yes

2. Has the statistical analysis been performed appropriately and rigorously? 

Reviewer #1: Yes

Reviewer #2: Yes

3. Have the authors made all data underlying the findings in their manuscript fully available?

Reviewer #1: Yes

Reviewer #2: Yes

4. Is the manuscript presented in an intelligible fashion and written in standard English?

Reviewer #1: Yes

Reviewer #2: Yes

5. Review Comments to the Author

Reviewer #1: This paper entitled “Influence of freeze-thaw and root combined action on soil mechanical characteristics and stability in the water-level-fluctuating zone of Baijiabao landslide in the Three Gorges Reservoir Area” discusses the strength variation of soil under the combined action of freeze-thaw and root system, and the resistance of root reinforcement to freeze-thaw deterioration is quantitatively expressed. The paper may be valuable for the civil engineering community. However, it needs some significant improvements before further processing.

Title: Title is very long. Please edit and revise the title and write more suitable title.

Abstract: The abstract should ideally include the main contributions and implications of the study. The abstract section should be improved considering the following structure: Introduction, problem statement, methodology, results, and conclusion.

Introduction: The introduction provides a clear and concise overview of the research topic, setting the stage for the study.

Please add a paragraph at the beginning of introduction about effect of natural effect such as freeze-thaw and dry-wet cycles on geotechnical parameters of geomaterials. Consider providing more context or background information to help readers understand the significance of the research problem.

The literature review needs to be completed, and it is not apparent what the novelty of this paper is. Your literature review should be updated. There are many new papers related to your research: https://doi.org/10.1061/(ASCE)MT.1943-5533.0003905, https://doi.org/10.1007/s10064-023-03427-6.

At the end of the Introduction, where the author introduces the central claim of his research or reasserts this claim, the paper should be including a paragraph to explain the importance of the subject, novelty, and originality of the paper.

materials and methods: Please write soil type based on USCS classification.

Please write in more details about the details of wave velocity test.

Test scheme: Please present a Table with more details instead of Table 2 for test program for showing details of tests.

On what basis have the number and temperatures of freeze-thaw cycles been chosen?

Results: in my opinion “results and dissection” is more suitable than “results and analysis” for section 3.

Ensure that all figures and tables are properly labeled and referred to in the text.

After summarizing your key findings, compare your results with previous studies.

The Manuscript should be polished by native speakers or a designated editing company to improve readability.

Some quantitative findings should support the conclusion. Please write the conclusion in more detail.

Reviewer #2: Your manuscript "Influence of freeze-thaw and root combined action on soil mechanical characteristics and stability in the water-level-fluctuating zone of Baijiabao landslide in the Three Gorges Reservoir Area"(PONE-D-23-42173) requires amendment. Although it is of interest, several details need to be clarified or corrected.

1. In line 37. “Compared to ordinary soil, the soil in the water-level-fluctuating zone has a substantially greater water content and the degradation in the soil of the water-level-fluctuating zone is more severe.” Is there a direct basis for this conclusion to be introduced as background? Citing relevant literature to strengthen the testimony is a good way to improve.

2. In line 78. “lt is a typical soil landslide induced by the failure of shallow soil strength caused by freeze-thaw action.” However, in line 82, the maximum freezing depth of soil is about 30 cm. Is the actual depth of the sliding belt only 30cm? It is recommended that this be explained in the text or in a diagram.

3. The gradation curve of soil sample in Fig.2 is incomplete. According to the standard of geotechnical test method, when the mass of the specimen with particle size less than 0.075mm is more than 10% of the total mass, the composition of the particles with particle size less than 0.075mm should be determined according to the densitometer method or pipette method.

4. In line 94. Previous pull-out test results are mentioned. However, no relevant literature citations or experimental results are given or presented.

5. According to Figure 4, the highest measured root content in the field was over 20%, while the experimental design was as high as 90%?

6. The layout of Table 2 is not aesthetically pleasing, and the parentheses try not to be misplaced.

7. Figure 5 is too redundant, flowcharts and diagrams of experimental procedures should be concise and clear, and detailed descriptions already in the text are not recommended to be repeated in flowcharts.

8. About Shear Strength Parameters. In general, fully saturated specimens of triaxial UU tests have only one effective stress Mohr's circle at the time of damage, and the envelope of the damage Mohr's circle for all the different enclosing pressures is a nearly horizontal straight line. It is therefore not possible to determine an effective shear strength parameters by measuring pore water pressure. It is usually only used to determine the undrained shear strength Cu. Therefore the angle of internal friction in Fig. 9(b) taken to 17° is needed to discussion. A detailed description of the triaxial test and shear strength parameters can be found in the soil mechanics textbook.

6. PLOS authors have the option to publish the peer review history of their article (what does this mean?). If published, this will include your full peer review and any attached files.

Reviewer #1: **Yes: **Meysam Bayat

Reviewer #2: No

---

## [Author Response · Author response to Decision Letter 0]

19 Mar 2024

Response to Reviewer Comments

Response to Academic Editor

Point 1: Please ensure that your manuscript meets PLOS ONE's style requirements, including those for file naming.

Response 1: We have now re-typed and checked the format in strict accordance with the manuscript template you gave. If there is still a problem of incorrect formatting, please give us instant feedback, and we will hand over to a professional editing and polishing company for typesetting after the manuscript content is determined to be unchanged.

Point 2: In your Methods section, please provide additional information regarding the permits you obtained for the work. Please ensure you have included the full name of the authority that approved the field site access and, if no permits were required, a brief statement explaining why.

Response 2: The research area ( Baijiabao landslide ) in this study is an open area, so no additional work permit is required.

Point 3: Please note that PLOS ONE has specific guidelines on code sharing for submissions in which author-generated code underpins the findings in the manuscript. In these cases, all author-generated code must be made available without restrictions upon publication of the work.

Response 3: We are able to provide unlimited data and code of this manuscript, we have uploaded all the relevant data of our chart, if there are other data need to contact us at any time.

Point 4: Please state what role the funders took in the study. If the funders had no role, please state: "The funders had no role in study design, data collection and analysis, decision to publish, or preparation of the manuscript."

Response 4: The funders had no role in study design, data collection and analysis, decision to publish, or preparation of the manuscript.

Point 5: We note that Figure 1 in your submission contain [map/satellite] images which may be copyrighted……

Response 5: According to your request, we have now deleted the satellite map in Figure 1 and only retained the engineering geological map of the landslide.

Point 6: We note that Figures 3, 5 and 13 in your submission contain copyrighted images……

Response 6: We have now replaced similar images in Figure 3, Figure 5 and Figure 13 and annotated them in the revised manuscript.

Response to Reviewer 1 Comments

Point 1: Title is very long. Please edit and revise the title and write more suitable title.

Response 1: Thank you for your suggestion , we realized that the title is really too specific, now we refer to similar articles, and the title changes more streamlined.

Point 2: The abstract should ideally include the main contributions and implications of the study. The abstract section should be improved considering the following structure: Introduction, problem statement, methodology, results, and conclusion.

Response 2: Thank you for your suggestion , We have now improved and supplemented the abstract according to the structure you suggested.

Point 3: Please add a paragraph at the beginning of introduction about effect of natural effect such as freeze-thaw and dry-wet cycles on geotechnical parameters of geomaterials. Consider providing more context or background information to help readers understand the significance of the research problem.

Response 3: Thank you for your suggestion , The background information we have given before may be too short. Now we supplement the description of the failure process of soil by freeze-thaw action and the significance of studying the soil in the water-level-fluctuating zone in the first paragraph of the introduction. The detailed revisions are shown in lines 41-53 of the revised manuscript.

Point 4: The literature review needs to be completed, and it is not apparent what the novelty of this paper is. Your literature review should be updated. 

Response 4: Thank you very much for your comments and kindly provided us with the literature closely related to our research. We have cited the literature you provided and supplemented more related literature.

Point 5: At the end of the Introduction, where the author introduces the central claim of his research or reasserts this claim, the paper should be including a paragraph to explain the importance of the subject, novelty, and originality of the paper.

Response 5: We have added a paragraph at the end of the introduction to emphasize the innovation and importance of our research according to your suggestions.

Point 6: Please write soil type based on USCS classification.

Response 6: Thank you for your suggestion, we have now given the soil type according to the USCS classification, as shown in the 106-107 lines of the revised manuscript.

Point 7: Please write in more details about the details of wave velocity test.

Response 7: Thank you for your suggestion, We have now given the instrument model used in the wave velocity test and the specific steps of the wave velocity test, as shown in lines 166-170 of the revised manuscript. We also supplemented the photos of the soil samples during the wave velocity test in Fig.5.

Point 8: Please present a Table with more details instead of Table 2 for test program for showing details of tests.

Response 8: We now modify and optimize Table 2.

Point 9: On what basis have the number and temperatures of freeze-thaw cycles been chosen?

Response 9: Thank you for your suggestion, The choice of temperature is based on the original text we have given, in the revised manuscript; the selection of the number of freeze-thaw cycles is based on our pre-test. When the number of freeze-thaw cycles exceeds 8 times, the strength of the sample basically does not decrease. We have now supplemented and explained it, as shown in the 152-165 lines of the revised manuscript.

Point 10: In my opinion “results and dissection” is more suitable than “results and analysis” for section 3.

Response 10: Thank you for your suggestion, We have revised the title of section 3 according to your suggestion.

Point 11: After summarizing your key findings, compare your results with previous studies.

Response 11: Thank you for your suggestion, We have added a comparison with other research results in the text in accordance with your recommendations.

Point 12: The Manuscript should be polished by native speakers or a designated editing company to improve readability.

Response 12: Thank you for your suggestion, We have now submitted the manuscript to the AJE official editing company for a comprehensive polishing.

Point 13: Some quantitative findings should support the conclusion. Please write the conclusion in more detail.

Response 13: Thank you for your suggestion, We have now added more quantitative data to the conclusion.

Response to Reviewer 2 Comments

Point 1: In line 37. “Compared to ordinary soil, the soil in the water-level-fluctuating zone has a substantially greater water content and the degradation in the soil of the water-level-fluctuating zone is more severe.” Is there a direct basis for this conclusion to be introduced as background? Citing relevant literature to strengthen the testimony is a good way to improve.

Response 1: Thank you for your suggestion , The soil in the water-level-fluctuating zone is formed in the water body and is in the underwater environment. It is immersed in water for a long time and has stronger water absorption. Therefore, compared with the soil on land, its water content is usually higher. We have now added the corresponding literature to prove this in accordance with your recommendations.

Point 2: In line 78. “lt is a typical soil landslide induced by the failure of shallow soil strength caused by freeze-thaw action.” However, in line 82, the maximum freezing depth of soil is about 30 cm. Is the actual depth of the sliding belt only 30cm? It is recommended that this be explained in the text or in a diagram.

Response 2: Thank you very much for pointing out our mistakes in language expression. The landslide is not entirely caused by freeze-thaw. What we want to express here is that the subsequent slip of the landslide after formation is mainly caused by freeze-thaw. Freezing and thawing and the root system mainly affect the changes in the shallow surface soil above the slide zone, thus indirectly leading to the subsequent sliding of the slope. The evidence is that the slip of the landslide in winter is much larger than that in other seasons. Obviously, our misrepresentation is misleading, and we have now made corresponding modifications. Thank you again for pointing out our mistakes, which is very important to us.

Point 3: The gradation curve of soil sample in Fig.2 is incomplete. According to the standard of geotechnical test method, when the mass of the specimen with particle size less than 0.075mm is more than 10% of the total mass, the composition of the particles with particle size less than 0.075mm should be determined according to the densitometer method or pipette method.

Response 4: Thank you for your suggestion ,We have now completed the gradation curve.

Point 4: In line 94. Previous pull-out test results are mentioned. However, no relevant literature citations or experimental results are given or presented.

Response 5: Thank you for your suggestion , We have added literature to confirm this.

Point 5: According to Figure 4, the highest measured root content in the field was over 20%, while the experimental design was as high as 90%?

Response 6: Fig.4 shows the relationship between underground root content and surface vegetation coverage. The maximum root content is 21.49 %, and the corresponding vegetation coverage is 90 %. Because the roots are buried underground and cannot be observed, we use vegetation coverage instead of root content, which can facilitate technicians to directly judge the reinforcement effect according to the degree of vegetation coverage on the surface. The relevant explanations are given in lines 137-146 of the revised manuscript.

Point 6: The layout of Table 2 is not aesthetically pleasing, and the parentheses try not to be misplaced.

Response 7: Thank you for your suggestion, We now modify and optimize Table 2.

Point 7: Figure 5 is too redundant, flowcharts and diagrams of experimental procedures should be concise and clear, and detailed descriptions already in the text are not recommended to be repeated in flowcharts.

Response 7: Thank you for your suggestion, We have now redrawn Figure 5 to make it more streamlined.

Point 8: About Shear Strength Parameters. In general, fully saturated specimens of triaxial UU tests have only one effective stress Mohr's circle at the time of damage, and the envelope of the damage Mohr's circle for all the different enclosing pressures is a nearly horizontal straight line. It is therefore not possible to determine an effective shear strength parameters by measuring pore water pressure. It is usually only used to determine the undrained shear strength Cu. Therefore the angle of internal friction in Fig. 9(b) taken to 17° is needed to discussion. A detailed description of the triaxial test and shear strength parameters can be found in the soil mechanics textbook.

Response 8: Thank you very much for your opinion. As you said, for saturated clay, the Mohr stress circle envelope obtained by UU test will approach the horizontal line, so the φ value cannot be obtained. For the unsaturated clay used in this paper, the UU test can not fully and accurately reflect the real mechanical behavior of unsaturated soil, and the obtained φ value is not accurate, so we think that your opinion is undoubtedly correct. In fact, the values of c and φ measured by different test methods ( UU, CU, CD ) are different. The UU test is carried out under undrained conditions. The volume of the sample is constant during the test, and the water content is constant. Changing the surrounding pressure increment will not change the effective stress in the sample, but only cause the change of pore water pressure. If the pre-shear consolidation pressure of the sample is large, the UU test will obtain a larger cohesion value and a smaller φ value. Under the action of consolidation pressure and pore water discharge, the spacing of soil particles in CU and CD tests is gradually shortened, the interaction between particles is strengthened, and the relative movement of particles is more difficult. Therefore, the measured φ value is usually larger than that of UU test. However, a large number of literatures have proved that the differences in the specific values of the parameters caused by different test methods usually do not affect the overall trend of the parameters when the same factor changes, so we believe that our conclusions are still reasonable. Thank you again for your suggestion, we think this clarification is very necessary.

---

## [Decision Letter · Decision Letter 1]

3 Apr 2024

Effect of freeze‒thaw cycles on root–soil composite mechanical properties and slope stability

PONE-D-23-42173R1

Dear Dr. Luo,

We’re pleased to inform you that your manuscript has been judged scientifically suitable for publication and will be formally accepted for publication once it meets all outstanding technical requirements.

Kind regards,

Shaker Qaidi

Academic Editor

PLOS ONE

Additional Editor Comments (optional):

Dear Authors,

I am pleased to inform you that your manuscript has been accepted for publication in our journal.

The reviewers acknowledged the importance of your work and found that it makes a significant contribution to the field. Your research methods were sound, the data supports the conclusions, and the paper is well-written overall.

Reviewers' comments:

Reviewer's Responses to Questions

**Comments to the Author**

1. If the authors have adequately addressed your comments raised in a previous round of review and you feel that this manuscript is now acceptable for publication, you may indicate that here to bypass the “Comments to the Author” section, enter your conflict of interest statement in the “Confidential to Editor” section, and submit your "Accept" recommendation.

Reviewer #1: All comments have been addressed

2. Is the manuscript technically sound, and do the data support the conclusions?

Reviewer #1: Yes

3. Has the statistical analysis been performed appropriately and rigorously? 

Reviewer #1: Yes

4. Have the authors made all data underlying the findings in their manuscript fully available?

Reviewer #1: Yes

5. Is the manuscript presented in an intelligible fashion and written in standard English?

Reviewer #1: Yes

6. Review Comments to the Author

Reviewer #1: The new version of the paper is wholly modified compared to the original version, and the article is acceptable for publication.

7. PLOS authors have the option to publish the peer review history of their article (what does this mean?). If published, this will include your full peer review and any attached files.

Reviewer #1: **Yes: **Meysam Bayat
